# Hydroxyzine Use and Mortality in Patients Hospitalized for COVID-19: A Multicenter Observational Study

**DOI:** 10.3390/jcm10245891

**Published:** 2021-12-15

**Authors:** Marina Sánchez-Rico, Frédéric Limosin, Raphaël Vernet, Nathanaël Beeker, Antoine Neuraz, Carlos Blanco, Mark Olfson, Cédric Lemogne, Pierre Meneton, Christel Daniel, Nicolas Paris, Alexandre Gramfort, Guillaume Lemaitre, Pedro De La Muela, Elisa Salamanca, Mélodie Bernaux, Ali Bellamine, Anita Burgun, Nicolas Hoertel

**Affiliations:** 1Département de Psychiatrie, Hôpital Corentin-Celton, AP-HP.Centre-Université de Paris, 92130 Issy-les-Moulineaux, France; frederic.limosin@aphp.fr (F.L.); cedric.lemogne@aphp.fr (C.L.); pedelamu@ucm.com (P.D.L.M.); nico.hoertel@yahoo.fr (N.H.); 2Department of Psychobiology & Behavioural Sciences Methods, Faculty of Psychology, Campus de Somosaguas Universidad Complutense de Madrid, 28223 Pozuelo de Alarcon, Spain; 3Institut de Psychiatrie et Neurosciences de Paris, Université de Paris, UMR_S1266, INSERM, 75014 Paris, France; 4UFR de Médecine, Faculté de Santé, Université de Paris, 75006 Paris, France; 5Hôpital Européen Georges Pompidou, Medical Informatics, Biostatistics and Public Health Department, AP-HP.Centre-Université de Paris, 75015 Paris, France; vernet.raphael@gmail.com; 6Unité de Recherche Clinique, Hopital Cochin, Assistance Publique-Hopitaux de Paris, 75004 Paris, France; nathanael.beeker@aphp.fr; 7Cordeliers Research Center, Université de Paris, UMRS 1138, INSERM, 75006 Paris, France; antoine.neuraz@aphp.fr (A.N.); anita.burgun@aphp.fr (A.B.); 8Department of Medical Informatics, Necker-Enfants Malades Hospital, AP-HP, Centre-Université de Paris, 75015 Paris, France; 9Division of Epidemiology, Services and Prevention Research, National Institute on Drug Abuse, 6001 Executive Boulevard, Bethesda, MD 20852, USA; Carlos.Blanco2@nih.gov; 10Department of Psychiatry, New York State Psychiatric Institute, Columbia University, 1051 Riverside Drive, Unit 69, New York, NY 10032, USA; mark.olfson@nyspi.columbia.edu; 11Laboratoire d’Informatique Médicale et d’Ingénierie des Connaissances en e-Santé, UMR 1142, INSERM, Sorbonne Université, Université Paris 13, 93017 Paris, France; pierre.meneton@spim.jussieu.fr; 12AP-HP, DSI-WIND (Web Innovation Données), 75184 Paris, France; christel.daniel@aphp.fr (C.D.); nicolas.paris@aphp.fr (N.P.); 13Laboratoire d’Informatique Médicale et d’Ingénierie des Connaissances en e-Santé, Sorbonne University, University Paris 13, Sorbonne Paris Cité, INSERM UMRS 1142, 75012 Paris, France; 14LIMSI, CNRS, Université Paris-Sud, Université Paris-Saclay, 91405 Orsay, France; 15Institut National de Recherche en Sciences et Technologies du Numérique (INRIA), Université Paris-Saclay, INRIA, CEA, 75012 Palaiseau, France; alexandre.gramfort@inria.fr (A.G.); guillaume.lemaitre@inria.fr (G.L.); 16Banque Nationale de Données Maladies Rares (BNDMR), Campus Picpus, Département WIND (Web Innovation Données), AP-HP, 75012 Paris, France; elisa.salamanca@aphp.fr; 17Direction de la Stratégie et de la Transformation, AP-HP, 75004 Paris, France; melodie.bernaux@aphp.fr; 18Unité de Recherche Clinique, Hôpital Cochin, AP-HP, Centre-Université de Paris, 75014 Paris, France; ali.bellamine@aphp.fr

**Keywords:** COVID-19, SARS-CoV-2, hydroxyzine, FIASMA, treatment, inpatients, mortality, death

## Abstract

(1) Background: Based on its antiviral activity, anti-inflammatory properties, and functional inhibition effects on the acid sphingomyelinase/ceramide system (FIASMA), we sought to examine the potential usefulness of the H1 antihistamine hydroxyzine in patients hospitalized for COVID-19. (2) Methods: In a multicenter observational study, we included 15,103 adults hospitalized for COVID-19, of which 164 (1.1%) received hydroxyzine within the first 48 h of hospitalization, administered orally at a median daily dose of 25.0 mg (SD = 29.5). We compared mortality rates between patients who received hydroxyzine at hospital admission and those who did not, using a multivariable logistic regression model adjusting for patients’ characteristics, medical conditions, and use of other medications. (3) Results: This analysis showed a significant association between hydroxyzine use and reduced mortality (AOR, 0.51; 95%CI, 0.29–0.88, *p* = 0.016). This association was similar in multiple sensitivity analyses. (4) Conclusions: In this retrospective observational multicenter study, the use of the FIASMA hydroxyzine was associated with reduced mortality in patients hospitalized for COVID-19. Double-blind placebo-controlled randomized clinical trials of hydroxyzine for COVID-19 are needed to confirm these results, as are studies to examine the potential usefulness of this medication for outpatients and as post-exposure prophylaxis for individuals at high risk for severe COVID-19.

## 1. Introduction

Global spread of the novel coronavirus SARS-CoV-2, the causative agent of coronavirus disease 2019 (COVID-19), has created an unprecedented infectious disease crisis worldwide [1,2,3,4]. The search for an effective treatment for patients with COVID-19 among all available medications is still urgently needed [5,6,7,8].

Antihistamines are widely used in the treatment of urticaria, allergic rhinitis, hay fever, conjunctivitis, and pruritus. They work by competitive binding to H1 receptors and inhibiting the action of histamine, a primary mediator of an early-phase allergic inflammatory response that also modulates the late-phase response characterized by cellular influx of eosinophils, neutrophils, basophils, mononuclear cells, and T lymphocytes [9]. In vivo and in vitro studies have also suggested additional anti-inflammatory properties of H1 antihistamines, including both receptor-dependent and receptor-independent mechanisms [10]. The receptor-dependent mechanisms may involve inhibition of NF-kB dependent cytokines (such as IL-1, IL-2, IL-6, IL-8, IL-12, TNF-α) [11] and adhesion proteins (such as ICAM-1, VCAM-1 and ECAM-1) [10]. The receptor-independent mechanisms, which require higher drug concentrations, may include inhibition of the release of pre-formed mediators, such as histamine and eosinophil proteins, by inflammatory cells as well as eicosanoid generation and oxygen free radicals’ production [10].

Prior research also supports in vitro antiviral effects of the H1 antihistamine hydroxyzine against MERS and hepatitis C virus [12], and recent studies suggest that antihistamine medications could interact with SARS-CoV-2 cellular entry and might be beneficial in reducing disease progression [13,14,15,16].

Among first generation antihistamines, hydroxyzine is one of the most prescribed antihistamines. Beyond its antihistaminic activity, hydroxyzine is also prescribed as a psychotropic medication for its tranquilizer and sedative properties, as it is a weak antagonist of the serotonin 5-HT2A, dopamine D2, and α1-adrenergic receptors. Prior work indicates several biological mechanisms that hydroxyzine may induce which could be beneficial against COVID-19.

First, hydroxyzine belongs to the group of functional inhibitors of acid sphingomyelinase (FIASMA) [17,18], and prior research supports that these molecules could be beneficial in COVID-19 patients, through potential subsequent antiviral and anti-inflammatory effects [7,8,17,19,20,21,22,23,24,25,26,27,28]. In addition to hydroxyzine, this group also comprises other medications commonly used in clinical practice, such as antidepressants (e.g., fluoxetine, fluvoxamine, amitriptyline), calcium channel blockers (e.g., amlodipine [18], bepridil), anti-Parkinson’s drugs (e.g., biperiden) and mucolytics (e.g., ambroxol [8,12,17]). In vitro and in vivo, these pharmacological compounds inhibit ASM, an enzyme that catalyzes the hydrolysis of sphingomyelin into ceramide and phosphorylcholine [20]. Preclinical data indicate that SARS-CoV-2 activates the ASM-ceramide system, resulting in the formation of ceramide-enriched membrane domains that facilitate viral entry and infection by clustering ACE2, the cellular receptor of SARS-CoV-2, and the release of pro-inflammatory cytokines [17,20,21]. The inhibition of the ASM/ceramide system by FIASMA antidepressants prevents infection of Vero E6 cells with SARS-CoV-2 [20]. Importantly, the reconstitution of ceramides in cells treated with these antidepressants restores the infection [20]. In healthy volunteers, oral use of the FIASMA antidepressant amitriptyline prevents infection of freshly isolated nasal epithelial cells, which is also restored after the reconstitution of ceramides in these cells [20]. In an observational multicenter retrospective study, use of a FIASMA medication upon hospital admission was associated with a substantially reduced risk of intubation or death [23]. Finally, plasma levels of certain sphingolipids, particularly sphingomyelins and ceramides, were found to correlate with disease clinical severity [29,30,31], inflammation markers [29,30] and viral load [30,31] in patients with COVID-19. Altogether, these data support the central role of ASM/ceramide system in SARS-CoV-2 infection that could be targeted by hydroxyzine, even if the magnitude of the in vitro FIASMA effect of this treatment (i.e., in vitro residual ASM activity of 43%) is slightly lower than that of fluoxetine (13%) and fluvoxamine (37.4%) [18].

Second, hydroxyzine binds with significant affinity to Sigma-1 receptors (S1Rs) [32], which have been shown to restrict the endonuclease activity of an endoplasmic reticulum stress sensor inositol-requiring enzyme 1 (IRE1) and to reduce cytokine expression without inhibiting classical inflammatory pathways [26,33,34].

Third, hydroxyzine is a medication with highly likely lysosomotropic behavior [35]. Lysosomotropic medications have shown antiviral effects on coronaviruses, likely due to interference with endosomal pathway [36], and are assumed to suppress the cytokine release syndrome and to attenuate the transition from mild to severe SARS-CoV-2 infection/COVID-19 [37,38].

Because prior research supports that severe COVID-19 is characterized by an excessive inflammatory response [39,40] and that viral load could be associated with worsening of symptoms [41], we hypothesized in this report that hydroxyzine could be beneficial in reducing mortality in inpatients with COVID-19. Short-term oral use of hydroxyzine is generally relatively well tolerated, although common side effects include sleepiness, headache, and dry mouth, and less common serious ones include delirium, QT prolongation, and torsade de pointes, particularly among older adults [1]. These risks can be significantly increased if this medication is administered parentally.

Observational studies of patients with COVID-19 taking medications for other indications can help decide which treatment should be prioritized for randomized clinical trials and minimize the risk of patient exposure to potentially harmful and ineffective treatments. To our knowledge, no prior clinical study has examined the potential usefulness of hydroxyzine in patients with COVID-19.

To this end, we used data from the Assistance Publique-Hôpitaux de Paris (AP-HP) Health Data Warehouse [23,42,43,44,45], which includes data on all patients who have been admitted for COVID-19 to any of 36 Greater Paris University hospitals.

In this report, we examined the association between hydroxyzine use at hospital admission and mortality among adult patients who have been admitted to these medical centers for COVID-19, while adjusting for potential confounders, including patients’ characteristics (such as sex, age, hospital, obesity, current smoking status), medical conditions, and use of other medications. We hypothesized that among patients hospitalized for COVID-19, hydroxyzine use would be associated with reduced mortality.

## 2. Materials and Methods

### 2.1. Setting and Cohort Assembly

A multicenter cohort study was conducted at 36 AP-HP hospitals from the beginning of the epidemic in France, i.e., 24 January until 1 May 2020 [23,42,43,44,45]. We included all adults aged 18 years or over who had been hospitalized in these medical centers for COVID-19. COVID-19 was ascertained by a positive reverse-transcriptase-polymerase-chain-reaction (RT-PCR) test on nasopharyngeal or oropharyngeal swab specimens.

This observational study using routinely collected data received approval from the Institutional Review Board of the AP-HP clinical Data Warehouse (decision CSE-20- 20_COVID19, IRB00011591). AP-HP clinical Data Warehouse initiatives ensure patient information and informed consent regarding the different approved studies through a transparency portal in accordance with European Regulation on data protection and authorization n°1980120 from National Commission for Information Technology and Civil Liberties (CNIL).

### 2.2. Data Sources

We used data from the AP-HP Health Data Warehouse (‘Entrepôt de Données de Santé (EDS)). This warehouse contains all the clinical data available on all inpatient visits for COVID-19 to any of 36 Greater Paris University hospitals. The data obtained included patients’ demographic characteristics, vital signs, laboratory test and RT-PCR test results, medication administration data, current medication lists, current diagnoses, and death certificates.

### 2.3. Variables Assessed

We obtained data for each patient at the time of the hospitalization regarding patients’ characteristics, other medications, medical indications of hydroxyzine prescription, and medical comorbidities. Patients’ characteristics included sex; age, which was categorized into four classes based on the OpenSAFELY study results [46] (i.e., 18–50, 51–70, 71–80, 81+); hospital, which was categorized into four classes following the administrative clustering of AP-HP hospitals in Paris and its suburbs based on their geographical location (i.e., AP-HP Centre—Paris University, Henri Mondor University Hospitals and at home hospitalization; AP-HP Nord and Hôpitaux Universitaires Paris Seine-Saint-Denis; AP-HP Paris Saclay University; and AP-HP Sorbonne University); obesity, which was defined as having a body mass index higher than 30 kg/m^2^ or an International Statistical Classification of Diseases and Related Health Problems (ICD-10) diagnosis code for obesity (E66.0, E66.1, E66.2, E66.8, E66.9); and self-reported current smoking status. Other medications included any medication prescribed according to compassionate use or as part of a clinical trial (i.e., hydroxychloroquine, azithromycin, remdesivir, tocilizumab, sarilumab, or dexamethasone) and any other antihistamine medication. To take into account medical indications of hydroxyzine prescription, we recorded whether patients had any current diagnosis, based on ICD-10 diagnosis codes recorded during the visit, of any anxiety, insomnia, nausea, urticaria or pruritus (F4, G47, R11, L50 or L29). Finally, medical comorbidities included: any other infectious diseases (A00-B99); neoplasms and diseases of the blood (C00-D89); mental disorders (F01-F99); diseases of the nervous system (G00-G99); cardiovascular disorders (I00-I99); respiratory disorders (J00-J99); digestive disorders (K00-K95); dermatological disorders (L00-L99); diseases of the musculoskeletal system (M00-M99); diseases of the genitourinary system (N00-N99); endocrine disorders (E00-E89); and eye-ear-nose-throat disorders (H00-H95). 

### 2.4. Hydroxyzine Use

Hydroxyzine use was defined as receiving this medication per os within the first 48 h from hospital admission. We used this delay because we considered that, in a context of overwhelming of all hospital units during the COVID-19 peak incidence, patients may not have received or been prescribed the treatment the first day of their admission. To minimize potential confounding effects of late prescription of hydroxyzine, patients who initiated this treatment more than 48 h after hospital admission were excluded from the analyses.

### 2.5. Outcome

Study baseline was defined as the date of hospital admission for COVID-19. The outcome was in-hospital all-cause mortality from study baseline until 1 May 2020.

### 2.6. Statistical Analysis

We calculated frequencies of all baseline characteristics described above in patients who received hydroxyzine at hospital admission and in those who did not, and compared them using standardized mean differences (SMD) [47].

To examine the associations between hydroxyzine use and mortality, we performed multivariable logistic regression models. Patients with a baseline prescription of hydroxyzine were compared with a reference group without any hydroxyzine prescription during the hospitalization. To help account for the nonrandomized prescription of hydroxyzine and reduce the effects of confounding, the primary analysis was a multivariable logistic regression that included sex, age, hospital, obesity, self-reported current smoking status, medical conditions, any medication prescribed according to compassionate use or as part of a clinical trial, and any other antihistamine medication.

As a sensitivity analysis, we performed a univariate logistic regression model in a matched analytic sample using a 1:1 ratio, based on the same variables used for the multivariable logistic regression analysis. In this analysis, to reduce the effects of confounding, optimal matching was used to obtain the smallest average absolute distance across all clinical characteristics between exposed patient and non-exposed matched controls. In case of non-balanced covariates, a multivariable logistic regression model adjusting for the non-balanced covariates was also performed.

We also performed five additional analyses. First, to increase our confidence that the results might not be due to unmeasured confounding or indication bias, we examined (i) this association among patients who received hydroxyzine only within the 3 months before hospital admission (and not during the visit) as compared to those who received it during the visit only, and (ii) the change of the magnitude of the effect of potential residual confounding on our results by varying the relationship of each potential confounder with mortality [48]. Second, we reproduced the main analyses among all patients hospitalized for COVID-19 while additionally adjusting by severity criteria at baseline. Severity of COVID-19 was defined as having at least one of the following criteria at hospital admission: respiratory rate >24 breaths/min or <12 breaths/min, resting peripheral capillary oxygen saturation in ambient air <90%, temperature >40 °C, systolic blood pressure <100 mm Hg, or high lactate levels >2 mmol/L [49]. Third, we reproduced the main analyses separately in patients admitted in ICUs and those admitted in normal wards. Fourth, we compared the mortality rate of patients who were prescribed hydroxyzine more than 48 h after admission with (i) those who received this medication within 48 h from hospital admission and (ii) those who never received this treatment during the visit. In the third and fourth analyses, we used a 1:2 ratio in the matched analytic sample to obtain adequate statistical power. Finally, we examined a potential dose-effect relationship by testing the association between the daily dose received (dichotomized at the median dose) with mortality within patients who received hydroxyzine at baseline.

For all associations, we performed residual analyses to assess the fit of the data, checked assumptions, and examined the potential influence of outliers [50]. Statistical significance was fixed a priori at two-sided *p*-value < 0.05. All analyses were conducted in R software version 3.6.3 (R Project for Statistical Computing).

## 3. Results

### 3.1. Characteristics of the Cohort

Of the 17,131 patients hospitalized for laboratory-confirmed COVID-19, 1963 patients (11.5%) were excluded because of missing data or their young age (i.e., less than 18 years old of age). Of the remaining 15,168 adult patients, 65 patients (0.4%) were excluded because they received hydroxyzine after more than 48 h after hospital admission. Of the remaining 15,103 adult inpatients, 164 (1.1%) used hydroxyzine within the first 48 h of hospitalization, with a mean delay between hospital admission and first prescription of 0.9 days (SD = 0.6; interquartile range (IQR) = 0–1.5)), at a median daily dose at baseline of 25.0 mg (SD = 29.5, IQR = 25.0–50.0 mg) (Figure 1).

RT-PCR test results were obtained after a median delay of 1.2 days (SD = 12.7; IQR = 0.6–7.2 days) from hospital admission date. This median delay was of 1.0 days in the exposed (SD = 12.9; IQR = 0.7–7.0 days) group, and of 1.2 days in the non-exposed (SD = 12.7; IQR = 0.6–7.2) group (Two-Sample Brown–Mood Median Test, Z = 1.83; *p* = 0.067). Over a mean follow-up of 14.5 days (SD = 17.9; median = 8 days; IQR = 1–24 days), 1589 patients (10.5%) died at the time of data cutoff on 1 May 2020. Among patients who received hydroxyzine medication at baseline, the mean follow-up was 14.4 days (SD = 15.4, median = 10 days; IQR = 5–18 days), while it was of 14.5 days (SD = 24.0, median = 8 days; IQR = 1–24 days) in those who did not (Welch Two Sample *t*-test, t = 0.9; *p* = 0.926). 

All patients’ characteristics, except for any other antihistamine medication, were significantly associated with death. A multivariable logistic regression model showed that age, sex, hospital, obesity, and most ICD-10 disorder categories, neoplasms and diseases of the blood, mental disorders, cardiovascular disorders, respiratory disorders, dermatological disorders, diseases of the genitourinary system, and eye-ear-nose-throat disorders were significantly and independently associated with mortality (Appendix A).

The distribution of patients’ characteristics according to hydroxyzine use is shown in Table 1. In the full sample, hydroxyzine use substantially differed according to all baseline characteristics (Table 1). The direction of these associations indicated older age and overall greater medical severity of patients receiving hydroxyzine than those who did not. In the matched analytic sample, these differences were substantially reduced and only sex, hospital, obesity and medication according to compassionate use or as part of a medical trial remained substantially different between groups (Table 1).

### 3.2. Study Endpoint

Among patients receiving hydroxyzine, death occurred in 18 patients (11.0%), while 1571 non-exposed patients (10.5%) had this outcome (Table 2). Despite the older age and the overall greater medical severity of patients receiving hydroxyzine at baseline than those who did not, the univariate unadjusted association between hydroxyzine use and mortality was not statistically significant (OR, 1.05, 95%CI, 0.64–1.72, *p* = 0.849) (Table 2). When taking into account differences in baseline characteristics between exposed and non-exposed patients, the multivariable logistic regression adjusting for age, sex, hospital, obesity, current smoking status, any medication prescribed according to compassionate use or as part of a clinical trial, any other antihistamine medication, and all medical conditions showed that hydroxyzine use was significantly associated with reduced mortality (AOR, 0.51; 95%CI, 0.29–0.88, *p* = 0.016) (Table 2; Figure 2).

In sensitivity analyses, the univariate logistic regression model in the matched analytic sample showed similar results as the multivariable logistic regression model (OR, 0.42; 95%CI, 0.22–0.80 *p* = 0.008) (Table 2).

Additional analyses indicated that patients who were prescribed hydroxyzine in the three months before but not during the visit were significantly at higher risk of death than those who received this medication only during the visit (Appendix A). This result suggests that the association between hydroxyzine use and reduced mortality is rather due to hydroxyzine use than to unmeasured characteristics of patients who were prescribed this medication, even if we cannot rule out that long-term side effects of prior use of this treatment might explain this finding. 

The quantitative bias analysis with observed imbalances showed fairly robust odds’ ratios under a wide range of assumed associations between potential confounders and the outcome. Associations for an apparent odds ratio of 0.51 are presented in Appendix A.

The association between hydroxyzine use and decreased mortality remained significant when additionally adjusting for clinical severity of COVID-19 at baseline (AOR, 0.49; 95%CI, 0.28–0.86, *p* = 0.013) (Appendix A). 

The association between hydroxyzine use and decreased mortality remained significant in both patients admitted to ICUs and those hospitalized in normal wards (Appendix A). However, the magnitude of the effect among patients admitted in ICUs should be considered with caution given the very limited numbers of patients taking hydroxyzine (*N* = 24) and events (*N* = 2) in this group. 

Taking hydroxyzine more than 48 h from hospital admission was significantly associated with increased mortality compared to taking hydroxyzine at baseline (i.e., within 48 h from hospital admission), and was not significantly different than not receiving this medication during the hospitalization (Appendix A). 

Finally, exposure to higher (median daily dose = 25 mg, IQR = 25.0–50.0 mg) than lower doses (median daily dose = 12.5 mg, IQR = 10.6–12.5 mg) of hydroxyzine at baseline was not significantly associated with mortality (Appendix A).

## 4. Discussion

In this multicenter retrospective observational study involving a relatively large sample of patients admitted to the hospital for COVID-19, our results suggest that hydroxyzine use at baseline, administered orally at a median daily dose of 25.0 mg (SD = 29.5) for a median duration of 14.4 days (SD = 15.4), was significantly and substantially associated with reduced mortality, independently of patients’ characteristics, other medications, medical indications of hydroxyzine prescription and medical comorbidities. Although these findings should be interpreted with caution due to the observational design of the study, they nonetheless provide support for conducting randomized double-blind placebo-controlled clinical trials with hydroxyzine against COVID-19.

Our study has several limitations. First, there are two possible major potential inherent biases in observational studies: unmeasured confounding and confounding by indication. Specifically, information on the precise indication for the prescription of hydroxyzine and whether the prescription was ongoing or first started at this time were not available. We tried to minimize the effects of confounding by hydroxyzine indication in different ways. First, we used a multivariable regression model to minimize the effects of confounding [35,36], taking into account main medical indications for hydroxyzine, including anxiety, insomnia, nausea, urticaria or pruritus, as well as a wide range of medical comorbidities, including any other infectious diseases, neoplasms and diseases of the blood, mental disorders, diseases of the nervous system, cardiovascular disorders, respiratory disorders, digestive disorders (K00-K95), dermatological disorders, diseases of the musculoskeletal system, diseases of the genitourinary system, endocrine disorders, and eye-ear-nose-throat disorders. Second, we performed a univariate Cox regression model as a sensitivity analysis in a matched analytic sample, that showed similar results. Third, although some amount of unmeasured confounding may remain, our analyses adjusted for numerous potential confounders and a quantitative bias analysis with observed imbalances suggested that residual confounding was unlikely to affect our results. Finally, the association was only observed in patients who received hydroxyzine within the first 48 h from hospital admission and not in those who received it only within the 3 months before hospital admission. Second, there were missing data for some baseline characteristic variables, including baseline clinical and biological markers of severity of COVID-19, which may be explained by the overwhelming of all hospital units during the COVID-19 peak incidence, and potential for inaccuracies in the electronic health records in this context. However, the associations observed between baseline characteristics and mortality are in line with prior epidemiological data [46]. Third, inflation of type I error might have occurred in secondary exploratory analyses due to multiple testing. Fourth, our results rely on a relatively limited number of patients who were prescribed hydroxyzine at baseline (N = 164), and limited statistical power may be responsible for the overestimation of the magnitude of the association [51]. Finally, despite the multicenter inpatient design, our results may not be generalizable to other settings or regions.

Multiple and possibly interrelated biological mechanisms may underlie this potential positive effect of hydroxyzine against COVID-19. First, hydroxyzine is a functional inhibitor of acid sphingomyelinase [17,18], and prior research supports that these molecules could be beneficial in COVID-19 patients, through potential subsequent antiviral and anti-inflammatory effects [7,8,17,19,20,21,22,23,24,25,26,27,28]. Second, hydroxyzine binds with significant affinity to Sigma-1 receptors (S1Rs) [32], which have been shown to restrict the endonuclease activity of an endoplasmic reticulum stress sensor inositol-requiring enzyme 1 (IRE1) and to reduce cytokine expression without inhibiting classical inflammatory pathways [26,34,35]. Finally, hydroxyzine is a medication with highly likely lysosomotropic behavior [36], and lysosomotropic medications have shown antiviral effects on coronaviruses, likely due to interference with endosomal pathway [36], and are assumed to suppress the cytokine release syndrome and to attenuate the transition from mild to severe SARS-CoV-2 infection/COVID-19 [37,38].

In this multicenter observational retrospective study involving patients hospitalized for COVID-19, hydroxyzine use was significantly and substantially associated with reduced mortality, independently of patients’ characteristics, other medications, medical indications of hydroxyzine prescription and medical comorbidities. Double-blind placebo-controlled randomized clinical trials of hydroxyzine for COVID-19 are needed to confirm these results, as are observational and clinical studies to examine the potential usefulness of this medication for outpatients with mild-to-moderate COVID-19 infection at high risk of progressing to severe COVID-19 and as post-exposure prophylaxis for individuals at high risk for severe COVID-19. 

## Figures and Tables

**Figure 1 jcm-10-05891-f001:**
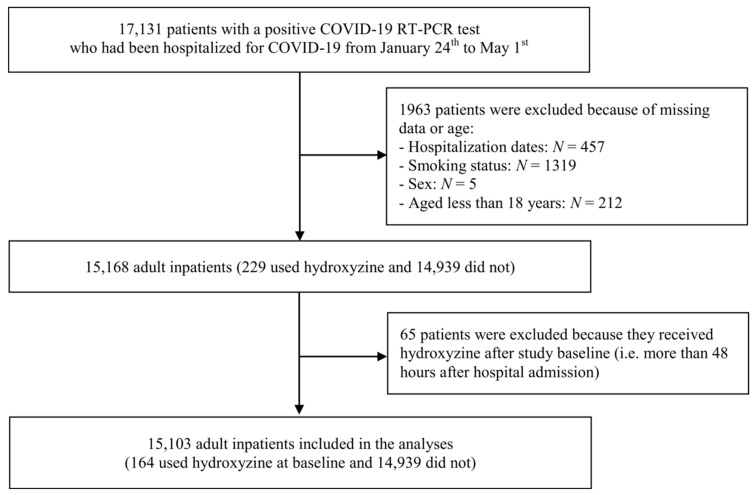
Study cohort.

**Figure 2 jcm-10-05891-f002:**
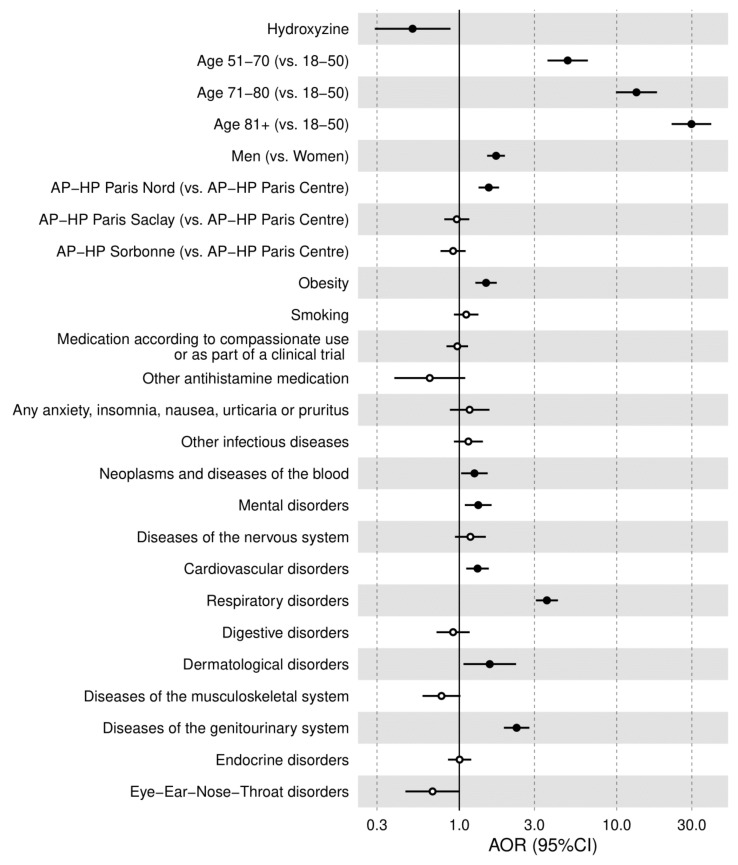
Multivariable logistic regression analysis for the association between hydroxyzine use and mortality in the full sample (*N* = 15,103). Dots in black indicate *p* < 0.05.

**Table 1 jcm-10-05891-t001:** Characteristics of inpatients with COVID-19 receiving or not receiving hydroxyzine.

	Exposed to Hydroxyzine (*N* = 164)	Not Exposed to Hydroxyzine (*N* = 14,939)	Non-Exposed Matched Group (*N* = 164)	Exposed to Hydroxyzine vs. Not Exposed to Hydroxyzine	Exposed to Hydroxyzine vs. Non-Exposed Matched Group
				Standardized mean differences	Standardized mean differences in the matched analytic sample
	*N* (%)	*N* (%)	*N* (%)	SMD	SMD
Patients’ characteristics					
Age				**0.300**	0.074
*18 to 50 years*	44 (26.8%)	5774 (38.7%)	39 (23.8%)		
*51 to 70 years*	65 (39.6%)	4716 (31.6%)	66 (40.2%)		
*71 to 80 years*	30 (18.3%)	1833 (12.3%)	32 (19.5%)		
*More than 80 years*	25 (15.2%)	2616 (17.5%)	27 (16.5%)		
Sex				**0.126**	**0.161**
*Women*	76 (46.3%)	7865 (52.6%)	63 (38.4%)		
*Men*	88 (53.7%)	7074 (47.4%)	101 (61.6%)		
Hospital				**0.557**	**0.147**
*AP-HP Centre—Paris University, Henri Mondor University Hospitals and at home hospitalization*	36 (22.0%)	6987 (46.8%)	45 (27.4%)		
*AP-HP Nord and Hôpitaux Universitaires Paris Seine-Saint-Denis*	56 (34.1%)	4059 (27.2%)	51 (31.1%)		
*AP-HP Paris Saclay University*	31 (18.9%)	1830 (12.2%)	33 (20.1%)		
*AP-HP Sorbonne University*	41 (25.0%)	2063 (13.8%)	35 (21.3%)		
Obesity ^a^				**0.253**	**0.105**
*Yes*	38 (23.2%)	2009 (13.4%)	31 (18.9%)		
*No*	126 (76.8%)	12,930 (86.6%)	133 (81.1%)		
Smoking ^b^				**0.194**	0.017
*Yes*	24 (14.6%)	1263 (8.5%)	23 (14.0%)		
*No*	140 (85.4%)	13,676 (91.5%)	141 (86.0%)		
Other medications					
Medication according to compassionate use or as part of a clinical trial ^c^				**0.482**	**0.107**
*Yes*	52 (31.7%)	1838 (12.3%)	44 (26.8%)		
*No*	112 (68.3%)	13,101 (87.7%)	120 (73.2%)		
Any other antihistamine medication				**0.276**	<0.001
*Yes*	10 (6.1%)	153 (1.0%)	10 (6.1%)		
*No*	154 (93.9%)	14,786 (99.0%)	154 (93.9%)		
Medical indications of hydroxyzine prescription					
Anxiety, insomnia, nausea, urticaria or pruritus				**0.276**	<0.001
*Yes*	14 (8.5%)	349 (2.3%)	14 (8.5%)		
*No*	150 (91.5%)	14,590 (97.7%)	150 (91.5%)		
Medical comorbidities					
Other infectious diseases ^d^				**0.347**	0.052
*Yes*	25 (15.2%)	740 (5.0%)	22 (13.4%)		
*No*	139 (84.8%)	14,199 (95.0%)	142 (86.6%)		
Neoplasms and diseases of the blood ^e^				**0.386**	0.048
*Yes*	30 (18.3%)	873 (5.8%)	27 (16.5%)		
*No*	134 (81.7%)	14,066 (94.2%)	137 (83.5%)		
Mental disorders ^f^				**0.371**	0.033
*Yes*	28 (17.1%)	824 (5.5%)	26 (15.9%)		
*No*	136 (82.9%)	14,115 (94.5%)	138 (84.1%)		
Diseases of the nervous system ^g^				**0.349**	0.017
*Yes*	23 (14.0%)	618 (4.1%)	24 (14.6%)		
*No*	141 (86.0%)	14,321 (95.9%)	140 (85.4%)		
Cardiovascular disorders ^h^				**0.372**	0.066
*Yes*	48 (29.3%)	2118 (14.2%)	53 (32.3%)		
*No*	116 (70.7%)	12,821 (85.8%)	111 (67.7%)		
Respiratory disorders ^i^				**0.787**	0.012
*Yes*	97 (59.1%)	3452 (23.1%)	98 (59.8%)		
*No*	67 (40.9%)	11,487 (76.9%)	66 (40.2%)		
Digestive disorders ^j^				**0.238**	0.041
*Yes*	15 (9.2%)	509 (3.4%)	17 (10.4%)		
*No*	149 (90.9%)	14,430 (96.6%)	147 (89.6%)		
Dermatological disorders ^k^				**0.206**	<0.001
*Yes*	7 (4.3%)	148 (1.0%)	7 (4.3%)		
*No*	157 (95.7%)	14,791 (99.0%)	157 (95.7%)		
Diseases of the musculoskeletal system ^l^				**0.292**	<0.001
*Yes*	15 (9.2%)	360 (2.4%)	15 (9.2%)		
*No*	149 (90.9%)	14,579 (97.6%)	149 (90.9%)		
Diseases of the genitourinary system ^m^				**0.291**	0.017
*Yes*	24 (14.6%)	879 (5.9%)	25 (15.2%)		
*No*	140 (85.4%)	14,060 (94.1%)	139 (84.8%)		
Endocrine disorders ^n^				**0.434**	0.064
*Yes*	53 (32.3%)	2148 (14.4%)	58 (35.4%)		
*No*	111 (67.7%)	12,791 (85.6%)	106 (64.6%)		
Eye-Ear-Nose-Throat disorders ^o^				**0.199**	<0.001
*Yes*	7 (4.3%)	160 (1.1%)	7 (4.3%)		
*No*	157 (95.7%)	14,779 (98.9%)	157 (95.7%)		

^a^ Defined as having a body-mass index higher than 30 kg/m^2^ or an International Statistical Classification of Diseases and Related Health Problems (ICD-10) diagnosis code for obesity (E66.0, E66.1, E66.2, E66.8, E66.9). ^b^ Current smoking status was self-reported. ^c^ Any medication prescribed as part of a clinical trial or according to compassionate use (e.g., hydroxychloroquine, azithromycin, remdesivir, tocilizumab, sarilumab or dexamethasone). ^d^ Assessed using ICD-10 diagnosis codes for certain infectious and parasitic diseases (A00-B99). ^e^ Assessed using ICD-10 diagnosis codes for neoplasms (C00-D49) and diseases of the blood and blood-forming organs and certain disorders involving the immune mechanism (D50-D89). ^f^ Assessed using ICD-10 diagnosis codes for mental, behavioral and neurodevelopmental disorders (F01-F99). ^g^ Assessed using ICD-10 diagnosis codes for diseases of the nervous system (G00-G99). ^h^ Assessed using ICD-10 diagnosis codes for diseases of the circulatory system (I00-I99). ^i^ Assessed using ICD-10 diagnosis codes for diseases of the respiratory system (J00-J99). ^j^ Assessed using ICD-10 diagnosis codes for diseases of the digestive system (K00-K95). ^k^ Assessed using ICD-10 diagnosis codes for diseases of the skin and subcutaneous tissue (L00-L99). ^l^ Assessed using ICD-10 diagnosis codes for diseases of the musculoskeletal system and connective tissue (M00-M99). ^m^ Assessed using ICD-10 diagnosis codes for diseases of the genitourinary system (N00-N99). ^n^ Assessed using ICD-10 diagnosis codes for endocrine, nutritional and metabolic diseases (E00-E89). ^o^ Assessed using ICD-10 diagnosis codes for diseases of the eye and adnexa (H00-H59) and diseases of the ear and mastoid process (H60-H95). SMD > 0.1 in bold indicate substantial differences. Abbreviation: SMD, standardized mean difference.

**Table 2 jcm-10-05891-t002:** Association between hydroxyzine use and mortality in the full sample and in the matched analytic sample.

	Number of Events/Number of Patients	Crude Logistic Regression Analysis	Multivariable Logistic Regression Analysis ^a^	Multivariable Logistic Regression Analysis ^b^	Number of Events/ Number of Patients	Univariate Logistic Regression in the Matched Analytic Sample (1:1)	Multivariable Logistic Regression Analysis in the Matched Analytic Sample (1:1) ^c^
	*N* (%)	OR (95%CI; *p*-value)	AOR (95%CI; *p*-value)	AOR (95%CI; *p*-value)	*N* (%)	OR (95%CI; *p*-value)	AOR (95%CI; *p*-value)
Hydroxyzine	18/164 (11.0%)	1.05 (0.64–1.72; 0.849)	0.53 (0.31–0.91; 0.023 *)	0.51 (0.29–0.88; 0.016 *)	18/164 (11.0%)	0.44 (0.24–0.81; 0.008 *)	0.42 (0.22–0.80; 0.008 *)
No hydroxyzine	1571/14,939 (10.5%)	Ref.	Ref.	Ref.	36/164 (22.0%)	Ref.	Ref.

^a^ Adjusted by age; sex; anxiety, insomnia, nausea, urticaria or pruritus; other infectious diseases; neoplasms and diseases of the blood; mental disorders; diseases of the nervous system; cardiovascular disorders; respiratory disorders; digestive disorders; dermatological disorders; diseases of the musculoskeletal system; diseases of the genitourinary system; endocrine disorders; and eye-ear-nose-throat disorders. ^b^ Adjusted by age; sex; hospital; obesity; smoking status; medication according to compassionate use or as part of a clinical trial; any other antihistamine medication; anxiety, insomnia, nausea, urticaria or pruritus; other infectious diseases; neoplasms and diseases of the blood; mental disorders; diseases of the nervous system; cardiovascular disorders; respiratory disorders; digestive disorders; dermatological disorders; diseases of the musculoskeletal system; diseases of the genitourinary system; endocrine disorders; and eye-ear-nose-throat disorders. ^c^ Adjusted by sex; hospital; obesity; and medication according to compassionate use or as part of a clinical trial. * *p*-value is significant (*p* < 0.05). Abbreviations: OR, odds ratio; AOR, adjusted odds ratio; CI, confidence interval.

## Data Availability

Data from the AP-HP Health Data Warehouse can be obtained upon request at https://eds.aphp.fr//, (accessed on 20 May 2020).

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
