# Peer review of "Hydroxyzine Use and Mortality in Patients Hospitalized for COVID-19: A Multicenter Observational Study"

_jcm, 2021, doi:10.3390/jcm10245891_

Round 1

Reviewer 1 Report

The paper of Sánchez-Rico et al. describes the potential benefit of hydroxyzine on the survival of hospitalized patients. I read the paper with great interest. Many therapeutic approaches are currently part of scientific investigations in the COVID-19 pandemic. An important aspect is that possibly already established and approved drugs, and their efficacy profile can still bring great benefits here, especially with regard to the cost of drugs. This can be seen from the numerous publications already available.

In the present study, the antihistamine hydroxyzine and its potential impact on mortality were investigated. Based on the results Sánchez-Rico et al. found reduced mortality of patients with prescription of hydroxyzine. Some remarks on the present manuscript must nevertheless be put forward for discussion in order to give the reader more clarity.

The indication for prescribing hydroxyzine does not come out clearly. Were these data also collected? Have any other drugs been collected in addition to the data? What does the mortality endpoint refer to, hospital mortality?

It might be a bit confusing for the reader to understand the many correlations and the resulting possible influencing factors. What role do the individual diseases play in the course of the COVID disease? Surely not all co-morbidities are causal for the course of the disease?

The severity of the disease related to the therapeutic effects would be beneficial. Are there any data on which patients were treated in normal wards or ICU? May hydroxyzine prevent against severe courses of COVID?

The discussion should be drawn more still into account to the obtain results. The discussion of possible causes for the presented results in relationship to effects of hydroxyzine are missing. This would be beneficial for the presented data.

What is the reason about possible therapeutic effects and or an explanation why just apparently an acute prescription, but no earlier prescription of hydroxyzine brings benefits?

Some limitations are discussed in detail. Nevertheless, the group size of patients who received hydroxyzine compared to the control cohort is also essential. Thus, may limited the data interpretation.

In addition to FIASMA, other approaches to the potential benefits of medication are possible. Interesting aspects also arise about lysosomotropic effects of drugs (PMID: 33692684).

Reviewer 2 Report

The introduction must state whether any other drugs which have a FIASMA action (for example biperiden) have been evaluated in a similar context. You did mention fluvoxamine but there are multiple other drugs which have this effect.

The risks of hydroxyzine use are not clearly presented, as the pro-arrythmic and QT prolonging effects of the drug pose a significant risk if it is administered parenterally.  Was,the drug administered per os in all patients?

Though I understand your rationale for excluding patients who where prescribed hydroxyzine more than 48 hours after admission, they should be included in multiple sensitivity analyses (compared with those who received the drug within 48 hours and those who were not exposed to it). 

Please state that the findings in no way endorse the prophylactic use of hydroxyzine. Despite its status as a prescription drug, in my country it can be bought without a prescription and the public may stockpile or use the drug in large quantities if these findings are sensationalized.

Though the association between hydroxyzine exposure is and reduced mortality is statistically significant, it may not be clinically significant as the estimated OR is slightly greater than 1.

Round 2

Reviewer 1 Report

Sánchez-Rico et al. addressed the main points raised form my previous review. Through reading the revised manuscript, I only have a few minor points and recommendations which should be addressed.

Certainly, it is important to address all possible confounder in context to the expected hypothesis to minimize the bias and to get robust results.  However, it would surely help the reader to have a more precise indication of the classification of the individual confounders, possibly also with regard to medical side effects or/and or other comorbidities.

Further, it would be helpful to discuss the reason for the obtain phenomenon whereas the positive effects of hydroxyzine while short prescription versus the negative effects while prior prescription. Would it be chronical side effects of longer duration of the prescription of the medication, or other side effects?

What might be the potential prophylactic effects of hydroxyzine in respect to disease progression and the results of prescription for longer duration and what might be the best moment to start the medication?

Are there any reasons for the results to disease severity and whereas the effect seems to be more pronounced to ICU patient than to patient to normal ward?

Author Response

December 10th, 2021,

Prof. Emmanuel Andrès, MD,

Editor-in-Chief, Journal of Clinical Medicine,

Dear Professor Andrès,

Thank you for the opportunity to submit a revised version of manuscript jcm-1491621-R1, “Hydroxyzine use and mortality in patients hospitalized for COVID-19: A multicenter observational study”. We want to thank the reviewers for their helpful comments, which have allowed us to strengthen this manuscript. A point-by-point response is included below, and the corresponding changes have been made in the manuscript and have been highlighted in yellow in a separate file.

The revised manuscript contains no data, patient information, or other material or results that have been published or are in press or submitted elsewhere.

We look forward to hearing from you and thank you in advance for considering this contribution.

Best Regards,

Marina Sánchez-Rico, MPH, and Nicolas Hoertel, MD, MPH, PhD,

On behalf of the authors and of AP-HP / Université de Paris / INSERM Covid-19 research collaboration / AP-HP Covid CDR Initiative / “Entrepôt de Données de Santé” AP-HP Consortium

Reviewer #1

1/ Sánchez-Rico et al. addressed the main points raised form my previous review. Through reading the revised manuscript, I only have a few minor points and recommendations which should be addressed.

Answer: We thank the reviewer for this positive comment.

2/ Certainly, it is important to address all possible confounder in context to the expected hypothesis to minimize the bias and to get robust results.  However, it would surely help the reader to have a more precise indication of the classification of the individual confounders, possibly also with regard to medical side effects or/and or other comorbidities.

Answer: We thank the reviewer for this comment and agree with her/him. For an easier presentation of the potential confounders we were able to include in our analyses, we classified them into four categories: patients’ characteristics, other medications, medical indications of hydroxyzine prescription, and medical comorbidities. This classification was included both in the method section and in Tables 1 and S1.

Following this comment, we modified the manuscript as follows:

P4: “We obtained data for each patient at the time of the hospitalization regarding pa-tients’ characteristics, other medications, medical indications of hydroxyzine prescription, and medical comorbidities. Patients’ characteristics included sex; age, which was categorized into 4 classes based on the OpenSAFELY study results [46] (i.e. 18-50, 51-70, 71-80, 81+); hospital, which was categorized into 4 classes following the administrative clustering of AP-HP hospitals in Paris and its suburbs based on their geographical loca-tion (i.e., AP-HP Centre – Paris University, Henri Mondor University Hospitals and at home hospitalization; AP-HP Nord and Hôpitaux Universitaires Paris Seine-Saint-Denis; AP-HP Paris Saclay University; and AP-HP Sorbonne University); obesity, which was defined as having a body mass index higher than 30 kg/m2 or an International Statistical Classification of Diseases and Related Health Problems (ICD-10) diagnosis code for obe-sity (E66.0, E66.1, E66.2, E66.8, E66.9); and self-reported current smoking status. Other medications included any medication prescribed according to compassionate use or as part of a clinical trial (i.e. hydroxychloroquine, azithromycin, remdesivir, tocilizumab, sarilumab, or dexamethasone) and any other antihistamine medication. To take into ac-count medical indications of hydroxyzine prescription, we recorded whether patients had any current diagnosis, based on ICD-10 diagnosis codes recorded during the visit, of any anxiety, insomnia, nausea, urticaria or pruritus (F4, G47, R11, L50 or L29). Finally, medical comorbidities included: any other infectious diseases (A00-B99); neoplasms and diseases of the blood (C00-D89); mental disorders (F01-F99); diseases of the nervous system (G00-G99); cardiovascular disorders (I00-I99); respiratory disorders (J00-J99); di-gestive disorders (K00-K95); dermatological disorders (L00-L99); diseases of the muscu-loskeletal system (M00-M99); diseases of the genitourinary system (N00-N99); endocrine disorders (E00-E89); and eye-ear-nose-throat disorders (H00-H95).”

3/ Further, it would be helpful to discuss the reason for the obtain phenomenon whereas the positive effects of hydroxyzine while short prescription versus the negative effects while prior prescription. Would it be chronical side effects of longer duration of the prescription of the medication, or other side effects?

Answer: We thank the reviewer for this comment. This additional analysis was performed to increase our confidence that this protective association between hydroxyzine and mortality was possibly due to the medication per se and not explained by unmeasured characteristics of patients who were prescribed hydroxyzine. The finding that patients with a past (and not current) prescription of hydroxyzine have increased risk of death compared to those currently taking this medication increases our confidence that this association is related to the use of hydroxyzine and not to characteristics of patients who were prescribed this medication. However, we agree with the reviewer that we cannot rule out that long-term side effects of prior use of this medication might explain this result.

Following this comment, we modified the manuscript as follows:

P11: “This result suggests that the association between hydroxyzine use and reduced mortality is rather due to hydroxyzine use than to unmeasured characteristics of patients who were prescribed this medication, even if we cannot rule out that long-term side effects of prior use of this treatment might explain this finding.” 

4/ What might be the potential prophylactic effects of hydroxyzine in respect to disease progression and the results of prescription for longer duration and what might be the best moment to start the medication?

Answer: Because this study is an observational study involving patients admitted to the hospital for COVID-19, we chose to prudently conclude that “In this multicenter retrospective observational study involving a relatively large sample of patients admitted to the hospital for COVID-19, our results suggest that hydroxyzine use at baseline, administered orally at a median daily dose of 25.0 mg (SD=29.5) for a median duration of 14.4 days (SD=15.4), was significantly and substantially associated with reduced mortality, independently of hydroxyzine use was significantly and substantially associated with reduced mortality, independently of patients’ characteristics, other medications, medical indications of hydroxyzine prescription and medical comorbidities”. Unfortunately, this study is unable to respond to the very important questions raised by the reviewer about treatment duration and ideal time to start this treatment to potentially observe a preventive or curative effect against COVID-19.

Following this comment, we modified the manuscript as follows:

P14: “In this multicenter observational retrospective study involving patients hospitalized for COVID-19, hydroxyzine use was significantly and substantially associated with reduced mortality, independently of patients’ characteristics, other medications, medical indications of hydroxyzine prescription and medical comorbidities. Double-blind placebo-controlled randomized clinical trials of hydroxyzine for COVID-19 are needed to confirm these results, as are observational and clinical studies to examine the potential usefulness of this medication for outpatients with mild-to-moderate COVID-19 infection at high risk of progressing to severe COVID-19 and as post-exposure prophylaxis for individuals at high risk for severe COVID-19. Double-blind placebo-controlled randomized clinical trials of hydroxyzine for COVID-19 are needed to confirm these results, as are observational and clinical studies to examine the potential usefulness of this medication for outpatients with mild-to-moderate COVID-19 infection at high risk of progressing to severe COVID-19 and as post-exposure prophylaxis for individuals at high risk for severe COVID-19.”

5/ Are there any reasons for the results to disease severity and whereas the effect seems to be more pronounced to ICU patient than to patient to normal ward?

Answer: We thank the reviewer for this comment. The association between hydroxyzine use and decreased mortality remained significant in both patients admitted to ICUs and those hospitalized in normal wards (Table S4). However, the magnitude of the effect among patients admitted in ICUs should be considered with caution given the very limited numbers of patients taking hydroxyzine (N=24) and events (N=2) in this group.

Following this comment, we specified this point in the manuscript as follows:

P11: “The association between hydroxyzine use and decreased mortality remained significant in both patients admitted to ICUs and those hospitalized in normal wards (Table S4). However, the magnitude of the effect among patients admitted in ICUs should be considered with caution given the very limited numbers of patients taking hydroxyzine (N=24) and events (N=2) in this group.”

Reviewer #2

6/ All comments I raised have been addressed sufficiently. It is substantially different but much improved compared to the original.

Answer: We thank the reviewer for this positive comment.

We warmly thank the reviewers for their important help in improving our manuscript.

Reviewer 2 Report

All comments I raised have been addressed sufficiently. It is substantially different but much improved compared to the original.

Author Response

(The authors gave the same response as above.)
